# The Preliminary Results for Evaluating Cocoa Butter’s Hepatoprotective Effects against Lipid Accumulation and Inflammation in Adult Male Rats Chronically Fed Ethanol

**DOI:** 10.3390/bioengineering9100526

**Published:** 2022-10-06

**Authors:** Hsiao-Yun Chang, Jiun-Rong Chen, Yi-Hsiu Chen, Qian Xiao, Ya-Ling Chen, Suh-Ching Yang

**Affiliations:** 1School of Nutrition and Health Sciences, College of Nutrition, Taipei Medical University, Taipei 11031, Taiwan; 2Research Center of Geriatric Nutrition, College of Nutrition, Taipei Medical University, Taipei 11031, Taiwan; 3Nutrition Research Center, Taipei Medical University Hospital, Taipei 11031, Taiwan; 4School of Gerontology and Long-Term Care, College of Nursing, Taipei Medical University, Taipei 11031, Taiwan

**Keywords:** alcoholic liver disease, lard, cocoa butter, saturated fatty acid, rat

## Abstract

The purpose of this study was to clarify the role of saturated fats from cocoa butter (plant source) compared with lard (animal source) on alcoholic liver damage in rats. Male Wistar rats were fed either a control diet (C) or an ethanol diet (E), and the dietary fats (corn oil, olive oil, and safflower oil) of these two diets were further replaced by lard (CL, EL) or cocoa butter (CC, EC). After 8-week feeding, plasma aspartate aminotransferase (AST) and alanine aminotransferase (ALT) activities, hepatic triglyceride (TG) levels, plasma intercellular adhesion molecular (ICAM)-1 levels, hepatic cytochrome P450 2E1 (CYP2E1) protein expression, and hepatic interleukin (IL)-1β significantly increased in the E group compared to the C group. In addition, hepatic histopathological scores of fatty changes, inflammatory cell infiltration, and degeneration and necrosis in the E group were significantly higher compared to those in the C group. However, fatty changes were significantly inhibited only in the EC group as well as hepatic inflammatory cell infiltration, degeneration, and necrosis being significantly lower in the EL and EC groups. Plasma ICAM-1 and hepatic tumor necrosis factor (TNF)-α, IL-1β, IL-6, and IL-10 levels were significantly lower in the EL and EC groups than those in the E group. Moreover, a correlation analysis showed that hepatic histopathological scores of degeneration and necrosis were significantly positively correlated with erythrocytic oleic acid (C18:1) and were negatively correlated with linoleic acid (C18:2). In conclusion, cocoa butter protected the liver against lipid accumulation and inflammation in rats chronically fed ethanol.

## 1. Introduction

Consuming excessive amounts of alcohol can aggravate liver damage, because lipids accumulate in the liver [1,2]. Several drugs are under development to reverse steatosis, but they are still in the early stages of development [3]. In addition to abstinence from alcohol, nutritional management of alcoholic hepatic steatosis plays an important role in treating all forms of alcoholic liver disease (ALD) [4].

It has been shown that the dietary fat content, bond lengths of carbon chains, the numbers of double bonds, and fatty acid compositions are associated with the development of ALD [5]. Several studies showed that diets enriched in saturated fatty acids (SFAs) or medium-chain triglycerides (MCTs) protected against alcoholic liver injury compared to diets containing polyunsaturated fatty acids (PUFAs) using either the Lieber–DeCarli liquid diet or an intragastric ethanol-fed animal model [6,7,8,9,10]. You et al. indicated that the protective effects of SFA-rich cocoa butter against alcoholic fatty liver may occur via the sirtuin (SIRT)-1-sterol regulatory element-binding protein 1 (SREBP-1) histone H3 axis, which suppresses expressions of genes encoding lipogenic enzymes and slows the synthesis of hepatic FAs [11]. Moreover, SFAs might increase adiponectin levels and upregulate peroxisome proliferator-activated receptor (PPAR)-α, peroxisome proliferator-activated receptor gamma coactivator (PGC)-1α, and adenosine monophosphate (AMP)-activated protein kinase (AMPK) expression which could increase FA oxidation and reduce lipid synthesis in the liver [12]. However, most of those studies used plant oils such as palm oil, cocoa butter, and MCTs as the only fat sources for the SFAs. In addition, there is little evidence about the comparison of plant SFAs with animal SFAs on the amelioration of ALD based on lipid metabolism, oxidative stress, and inflammatory response. On the other hand, saturated fats were also shown to increase the risk of cardiovascular diseases (CVDs) and induce insulin resistance [13,14]. Debate is still ongoing regarding the true overall long-term beneficial cardiovascular effects of SFAs [15].

Cocoa butter, a fat derived from cocoa plants and predominantly found in dark chocolate contains an average of 33% oleic acid (18:1 monounsaturated), 25% palmitic acid (16:0 saturated), and 33% stearic acid (18:0 saturated) [16]. Though it is generally considered that saturated fats overall adversely increase the total cholesterol and low-density lipoprotein cholesterol levels, early studies have also suggested stearic acid may be non-cholesterolemic [17,18,19]. 

On the other hand, lard (rich in palmitic acid) is the most common source of dietary fat in Taiwan. Taiwanese often use lard to fry vegetables and make Chinese-style desserts. Our previous study found that lard inhibited fat accumulation but induced fibrosis in rats fed an ethanol-containing diet [20]. However, the association between the blood FA composition and the severity of liver damage was not discussed in previous studies [6,7,8,9,10,20].

Considering the above issues, we hypothesized that plant- and animal-derived sources of SFAs would have different effects on alcoholic liver injury due to different FA compositions, and the erythrocytic FA composition would be closely correlated with the severity of alcoholic liver damage. This study was carried out to explore the proposed hypotheses focusing on the pathogenic mechanisms of ALD in rats fed an alcoholic liquid diet containing cocoa butter (a plant-derived fat) or lard (an animal-derived fat).

## 2. Materials and Methods

### 2.1. Animals and Study Protocol

Sixty aged 8-week-old male Wistar rats (National Laboratory Animal Center, Taipei, Taiwan), weighing 250–280 g, were used in this study. All rats were kept in a room at 55 ± 10% humidity and 23 ± 2 °C. All procedures were approved by the Institutional Animal Care and Use Committee of Taipei Medical University (with identification code LAC-2013-0193). After 1 week of acclimation, rats were divided into six groups (10 rats/group) according to aspartate aminotransferase (AST) and alanine aminotransferase (ALT) activities as liver function indicators such that no differences in liver damage existed among the groups before the experiment. Rats were fed either an ethanol liquid diet (E group) or an isocaloric pair-feeding control liquid diet (C group) modified from the Lieber–DeCarli formula with a Ritcher drinking tube [21]. Then, both the ethanol and control liquid diets were further replaced with lard for rats in the CL and EL groups, or cocoa butter for rats in the CC and EC groups. Dietary ingredients and FA compositions are respectively shown in Table 1 and Table 2. After 8 weeks of feeding, all rats were anesthetized and sacrificed. Blood samples were collected via the abdominal aorta, transferred to the EDTA-coated vacutainers, and centrifuged at 1200× *g* and 4 °C for 15 min. After removing plasma, the erythrocyte layer was immediately washed three times with an equal volume of 0.9% saline. In addition, livers were infused with a 0.9% saline solution from the portal vein and removed. All samples were stored at −80 °C until further analysis.

### 2.2. Liver Function Indicators

AST and ALT activities were analyzed with the ADVIA 1800^®^ Clinical Chemistry System (Siemens Healthcare, Erlangen, Germany).

### 2.3. Histopathological Examinations

Liver tissues were fixed in a 10% formaldehyde solution and embedded in paraffin. Liver paraffin sections were cut and stained with hematoxylin and eosin (H&E) and Masson’s trichrome stains. Experienced pathologists semi-quantitatively evaluated liver specimens according to the degree of tissue damage and fibrosis using a method described by Chiu et al. [22].

### 2.4. Lipid Profiles

Plasma triglycerides (TGs), total cholesterol (TC), high-density lipoprotein-cholesterol (HDL-C), and low-density lipoprotein-cholesterol (LDL-C) were measured with an ADVIA^®^ 1800 Chemistry Analyzer (Siemens Healthcare). Hepatic lipids were extracted based on Folch’s method [23], and TG and TC levels were analyzed by commercial kits (TR 213 and CH 7945, Randox Laboratories, Antrim, UK).

### 2.5. Oxidative Stress

Liver samples were homogenized in ice-cold buffer (150 mM NaCl, 50 mM Tris-HCl, 0.1% sodium dodecylsulfate (SDS), and 1% Triton X-100 at pH 7.2 and 4 °C) and centrifuged at 3000× *g* and 4 °C for 15 min, then the supernatant was used for analysis by Tietze’s [24] and Griffith’s methods [25].

The reduction in the hepatic glutathione (GSH)/oxidized glutathione (GSSG) ratio was measured as a marker of antioxidative nutrients. The GSH/GSSG ratio was calculated as (total GSH – 2GSSG)/GSSG. As a measure of lipid peroxidation, hepatic thiobarbituric acid-reactive substance (TBARS) levels were evaluated with a commercial kit (TBARS 10009055 (TCA Method) Assay Kit, Cayman Chemical, Ann Arbor, MI, USA) according to the manufacturer’s instructions.

### 2.6. Hepatic Cytochrome P450 2E1 (CYP2E1)

Liver tissues were homogenized in iced-homogenized buffer (0.25 M sucrose, 10 mM Tris-HCl, and 0.25 mM phenylmethylsulfonyl fluoride at 4 °C) and centrifuged at 17,000× *g* for 20 min to collect microsome-containing supernatants and then centrifuged at 1.05 × 10^5^× *g* and 4 °C for 60 min. Microsomal pellets were redissolved in suspension buffer (1 mM ethylenediaminetetraacetic acid, 1 mM dithiothreitol, and 50 mM Tris-HCl) with 0.1% of a protease inhibitor. CYP2E1 in liver tissues was quantified using a Western blot method based on procedures described by Chen et al. [26] and Yuan et al. [27].

### 2.7. Proinflammatory Markers

Plasma vascular cell adhesion protein (VCAM)-1 levels (SEA547Ra, Uscn Life Science, Wuhan, China), intercellular adhesion molecule (ICAM)-1 levels (Rat ICAM-1/CD54 Quantikine ELISA Kit, R&D Systems, Minneapolis, MN, USA), and E-selectin levels (SEA029Ra, Uscn Life Science) were determined by enzyme-linked immunosorbent assay (ELISA) kits, and the procedures followed the manufacturer’s instructions.

Hepatic cytokine levels including TNF-α, IL-1β, IL-6, and IL-10 were measured by a next-generation multiplex platform based on xMAP technology [28]. Liver homogenates (by the same method as the GSH/GSSG ratio) were mixed with 50 μL of antibody-coupled beads. Fifty microliters of supernatant was mixed with beads and reacted at room temperature for 60 min. After washing three times, biotinylated detection antibodies were added and reacted at room temperature for 30 min. Streptavidin-phycoerythrin was added and reacted at room temperature for 10 min. After washing three times, beads were suspended in assay buffer, read on a Bio-Plex suspension array system (Bio-Rad, Hercules, CA, USA), and then analyzed by Bio-Plex Manager software.

### 2.8. Fatty Acid Composition

Crude lipids of erythrocytes were extracted according to the method of Folch et al. [23]. Details of the FA methyl ester (FAME) analysis using capillary gas chromatography (GC) were the same as those described by Lee et al. [29]. The FA composition was analyzed by trace GC with a flame-ionized detector (ThermoQuest, Hudson, FL, USA) and an Rtx^®^-2330 column (30 m, 0.32 mm ID, 0.32 μm df, cat. #10724; Restek, Bellefonte, PA, USA). FA profiles of dietary fat and erythrocytes were identified according to retention times of appropriate FAME standards. Data were analyzed by Chrom-Card software (Thermo Fisher Scientific, Waltham, MA, USA) [30]. Composition data are expressed as weight percentages of total FAs.

### 2.9. Statistical Analysis

Data are presented as the mean ± standard error of the mean (SEM). SAS software vers. 9.1 (SAS Institute, Cary, NC, USA) and a one-way analysis of variance (ANOVA), followed by Fisher’s least significant difference (LSD) test were used to determine statistical differences among groups. Pearson’s correlation coefficient was used to determine the correlation between the erythrocytic fatty acid composition and the liver histological score. Statistical significance was assigned at the *p* < 0.05 level.

## 3. Results

### 3.1. Food Intake, Alcohol Intake, Body Weight (BW), and Liver Weight

Daily average food intake, alcohol intake, BWs, and relative liver weights in each group are shown in Table 3. There were no differences in daily average food intake or alcohol intake among the groups. The E group had a significantly lower BW than the C group; in addition, the BWs of the EL and EC groups were significantly lower compared to the E group. On the other hand, the E group had significantly higher relative liver weights compared to the C group, whereas no difference was found among the three alcohol-fed groups.

### 3.2. Liver Damage

#### 3.2.1. Liver Function Index

AST and ALT activities are shown in Table 4. After 8 weeks of feeding, AST and ALT activities had significantly increased in the E groups compared to the C group, whereas no difference was found among the E, EL, and EC groups.

#### 3.2.2. Liver Histopathological Examinations

Histopathological scores and features are shown in Table 5, Figure 1 and Figure 2. Compared to the C group, the E group had significantly higher fatty change, inflammatory cell infiltration and degeneration, and necrosis. However, the EL group showed the significantly lower fatty changes as well as the EL and EC groups having significantly lower inflammatory cell infiltration, degeneration, and necrosis compared to the E group. In addition, no difference was found in hepatic fibrosis among the groups.

### 3.3. Lipid Profiles 

Plasma lipid profiles are shown in Table 6. There was no change in plasma TG levels among all groups. Compared to the C group, only the plasma LDL-C level was significantly lower in the E group. However, plasma TC, HDL-C, and LDL-C levels were significantly higher while the TC/HDL-C ratio was significantly lower in the EL and EC groups than those in the E group.

The hepatic TC and TG levels are shown in Table 7. The E group had significantly higher hepatic TG levels compared to the C group and exhibited no differences compared to the EL and EC groups. There were no differences in hepatic TC levels among all groups.

### 3.4. Oxidative Stress

No significant differences in the hepatic GSH/GSSG ratio or TBARS level were found among all groups (Table 8). CYP2E1 protein expression was significantly elevated in the E group compared to the C group. However, no change in CYP2E1 protein expression was found among the E, EL, and EC groups (Figure 3).

### 3.5. Inflammatory Response

Plasma VCAM-1, ICAM-1, and E-selectin levels are shown in Table 9. No significant changes were observed in plasma VCAM-1 and E-selectin levels, whereas the plasma ICAM-1 level was significantly higher in the E group compared to the other groups.

Hepatic cytokine levels are shown in Table 10. Compared to the C group, only the hepatic IL-1β level was significantly increased in the E group, while it was significantly reduced in the EL and EC groups. Additionally, hepatic TNF-α, IL-1β, IL-6, and IL-10 levels were significantly decreased in the EL and EC groups compared to the E group.

### 3.6. Erythrocytic FA Composition

Table 11 presents the erythrocytic FA composition in each group. After alcohol feeding for 8 weeks, the E group showed significantly higher stearic acid (C18:0) and lower linoleic acid (C18:2) and arachidic acid (C20:0) in erythrocytes compared to the C group. However, compared to the E group, myristic acid (C14:0), palmitic acid (C16:0), palmitoleic acid (C16:1), and linoleic acid (C18:2) were significantly increased, while eicosadienoic acid (C20:2) was significantly decreased in the EL group. The EC group exhibited significantly higher stearic acid (C18:0) and linoleic acid (C18:2) and presented significantly lower palmitoleic acid (C16:1) than the E group. Comparing differences between the EL and EC groups, myristic acid (14:0), palmitic acid (C16:0), and palmitoleic acid (C16:1) were significantly decreased and stearic acid (C18:0) was significantly increased in the EC group.

Compared to the E group, the EL group had significantly higher total SFAs but lower total PUFAs which might have caused the higher SFA/PUFA and SFA/unsaturated FA (USFA) ratios. In addition, total SFAs were significantly increased in the EC group, leading to the higher SFA/monounsaturated FA (MUFA), SFA/PUFA, and SFA/USFA ratios.

### 3.7. Correlations between Erythrocytic FA Compositions and Hepatic Damage Scores

Oleic acid (C18:1), total MUFAs, total n-9, and the n-9/n-6 ratio were positively correlated with the degeneration and necrosis scores (Table 12). On the contrary, linoleic acid (C18:2) and the SFA/MUFA ratio were inversely correlated with the degeneration and necrosis scores (Table 12).

## 4. Discussion

### 4.1. Food Intake, Alcohol Intake, and BW

In this study, rats fed the ethanol liquid diet (the E, EC, and EL groups) showed significantly lower BWs, although their food intake was similar to rats fed the control liquid diet (the C, CL, and CC groups) (Table 3). This phenomenon was consistent with our previous studies [31,32]. Malnutrition is the most frequent complication in patients with *ALD*, including a low lean body mass, low plasma albumin, etc. [33]. Chronic alcohol abuse devastates gastrointestinal function, such as inhibiting bile acid secretion which reduces absorption of lipid- and fat-soluble vitamins, causing intestinal swelling and decreasing the activities of intestinal enzymes which cause malabsorption [34]. On the other hand, the EL and EC groups had significantly lower BWs compared to the E group (Table 3). Previous ALD-related animal studies showed similar results to this study [10,35]. However, as far as nutritional knowledge, SFAs are likely more obesogenic than MUFAs and PUFAs [36]. Relationships among SFAs, alcohol intake, and BW need to be further examined in future studies.

The average alcohol intake of rats in the E, EL and EC groups was around 10 g/kg BW/day (2 g/day), and this is the equivalent of 46 g/day in a human. A heavy alcohol intake is more than 50–60 g/day; 31–50 g/day is considered moderate, 21–30 g/day is considered mild, whereas 1–20 g/day is considered minimal [37]. Therefore, the ethanol intake of rats in this study may be considered as being comparable to that of a moderate drinker.

### 4.2. SFAs, Alcohol Intake, and Liver Damage

Significantly higher plasma AST and ALT activities (Table 4) as well as fatty changes, inflammatory infiltration, degeneration, and necrosis were found in the E group (Table 5, Figure 1). That is to say, alcoholic liver damage was successfully established in this study. Compared to the E group, both the EL and EC groups showed significantly lower hepatic cytokines and histopathological scores of inflammatory infiltration, degeneration, and necrosis (Table 3, Figure 1). You et al. reported that USFAs contained in the Lieber–DeCarli ethanol liquid diet were the key in inducing liver damage [11]. Linoleic acid which is rich in the Lieber–DeCarli ethanol liquid diet can be oxidized into oxidized linoleic acid metabolites (OXLAMs) and combined with transient receptor potential vanilloid 1 (TRPV1) in the liver, to then increase intracellular calcium levels [38]. Afterward, the activated pathway causes inflammatory cytokine synthesis, which triggers liver inflammation [39]. Therefore, it was speculated that replacing corn oil, olive oil, and safflower oil with lard or cocoa butter might diminish hepatic inflammation in rats fed an ethanol liquid diet.

### 4.3. SFAs, Alcohol Intake, and Lipid Metabolism

As shown in Table 6 and Table 7, the E group had significantly lower plasma LDL-C and higher hepatic TG contents compared to the C group. Very-low-density lipoprotein (VLDL) is assembled in the liver from TGs, cholesterol, and apolipoproteins and is converted to LDL and intermediate-density lipoprotein (IDL) by lipoprotein lipase in the bloodstream [39]. Previous studies found that chronic alcohol intake inhibited VLDL synthesis in the liver, which caused lipid accumulation in the liver because TGs could not be transported out of the liver [40,41]. Li et al. reported that chronic alcohol consumption stimulates lipolysis and promotes the release of free FAs from adipose tissues [42]. Then, free FAs enter the liver via the blood circulation and activate glycerol-3-phosphate acyltransferase 3 (GPAT3) which increases TG synthesis [38]. On the other hand, alcohol also elevates the SREBP-1c protein and reduces the PPAR-α protein which inhibits FA oxidation [42]. Therefore, it was speculated that chronic alcohol intake not only inhibits lipoprotein synthesis but also increases TG synthesis, which lead to lipid accumulation in the liver.

In this study, rats fed lard or cocoa butter under chronic alcohol intake presented higher TC, HDL-C, and LDL-C levels as well as a lower TC/HDL-C ratio compared to the E group (Table 6). The blood LDL-C concentration is determined by the rate of transformation from VLDL-C and the metabolic rate of the LDL-C and LDL receptor combination [43]. LDL receptors, which determine the blood LDL-C amount, can be activated by dietary FAs [44,45]. Siri-Tarino et al. reported that SFAs, such as lauric acid, myristic acid, and palmitic acid, suppress LDL receptor activity, resulting in reduced LDL clearance which leads to higher blood LDL-C levels [46]. In this study, lard and cocoa butter contained higher levels of palmitic acid, which might explain why rats in the EL and EC groups had higher plasma LDL-C levels. In addition, it was also indicated that SFAs increase HDL-C, but the TC/HDL-C ratio (a risk marker for CVD) is not altered [47]. In this study, higher HDL-C levels and lower TC/HDL-C ratios were observed in the EL and EC groups. Therefore, it was speculated that the decrease in the TC/HDL-C ratio in the EL and EC groups could be attributed to the effects of alcohol intake on lipoprotein biosynthesis.

However, there were no changes in hepatic fatty changes or TG levels in the EL and EC groups (Figure 1, Table 7). It was reported that using lard as a fat source (with fat providing 40% of total calories) with an ethanol diet elevated the plasma adiponectin level and hepatic PPAR-α expression which promoted FA oxidation and inhibited hepatic lipid accumulation [20]. Another study pointed out that cocoa butter increased the plasma adiponectin level and inhibited acetyl-CoA, a key enzyme in lipid synthesis, and then inhibited fatty liver in Sprague–Dawley rats fed an ethanol diet (with alcohol providing 27.5% of total calories) [9]. Thus, different animal strains, experimental periods, and total calories provided by saturated fat and ethanol in the diet are possible explanations of why the present study and previous studies showed different levels of liver lipid accumulation.

### 4.4. SFAs, Alcohol Intake, and Oxidative Stress

Chronic alcohol consumption increases ROS formation which is attributed to the hepatic cytochrome P-450, especially ethanol-inducible CYP2E1 in the microsomal ethanol-oxidizing system (MEOS). Therefore, CYP2E1 protein expression is commonly used as an indicator of oxidative stress induced by alcohol intake and is related to the hepatic pathological condition [48]. A previous study reported that when rats were given a CYP2E1 inhibitor, either diallyl sulfide or chlormethiazole, hepatic lipid peroxidation was inhibited, and alcoholic liver damage was extenuated [49]. It was indicated that mice with *CYP2E1* gene knock-off had lower liver lipid accumulation induced by chronic alcohol feeding compared to wild-type (WT) mice [50,51]. In this study, CYP2E1 protein expression was significantly higher in the E group, which showed that oxidative stress occurred in rats under chronic alcohol feeding (Figure 3). However, in this study, there were no differences between the E and C groups in the hepatic GSH/GSSG ratios or malondialdehyde (MDA) levels (Table 8). Compared to healthy individuals, alcoholics had lower erythrocytic GSH levels, glutathione synthetase activation, and higher lipid peroxidation and MDA [52]. Rats fed the Lieber–DeCarli liquid diet or given an ethanol solution by gastric gavage (8 g/kg BW) for 16 weeks presented lower hepatic mitochondria GSH levels and elevated lipid peroxidation [53]. It was surmised that the duration of alcohol feeding in the current study was too short to induce severe oxidative stress in the liver. Moreover, other biomarkers of oxidative stress, such as antioxidative enzymes, ROS detection, F2-isoprostanes, etc., should be analyzed in future research.

In this study, no changes in hepatic CYP2E1 protein expression, the GSH/GSSG ratio, or TBARS levels were observed among the E, EL, and EC groups (Figure 3, Table 8). In a high-fat diet-induced obese mice model, cocoa butter as dietary fat (40% of total calories) increased hepatic CYP2E1 messenger (m)RNA expression [54]. When using an ethanol diet with saturated FAs (with a ratio of beef tallow to MCT oil of 18:82), hepatic CYP2E1 protein expression, the GSH/GSSG ratio, and TBARS levels did not change [10]. Chen et al. used long-chain SFAs made of soy glyceride (with a ratio of C16:0, C18:0, and C18:1 of 12:85:3) as the source of dietary fat in an ethanol diet for mice, and hepatic CYP2E1 protein expression did not change, while TBARS levels significantly decreased [55]. In this study, lard and cocoa butter were used which are rich in C16:0 and C18:0 and belong to long-chain SFAs, but the ratios of C16:0, C18:0, and C18:1 differed from the previous study [55]. Thus, it was speculated that not only the length but also the ratio of SFAs affect the extent of alcohol-induced oxidative stress, including CYP2E1, GSH/GSSG, and TBARSs, but further clarification is needed.

### 4.5. SFAs, Alcohol Intake, and Inflammation

In this study, the E group had significantly higher plasma ICAM-1 and hepatic IL-1β levels compared to the C group (Table 9 and Table 10). This result showed that alcoholic hepatitis had occurred in rats, which was consistent with results of the hepatic pathological examination (Table 5, Figure 1) and our previous studies [16,18]. In a clinical study, it was found that plasma ICAM-1 levels were positively correlated with the severity of ALD [56]. Mice with ICAM-1 gene knock-off showed mild hepatic fat accumulation and inflammation compared to WT mice, so ICAM-1 was involved in ALD pathogenesis due to induction of neutrophil chemotaxis in hepatocytes [57]. As for the higher hepatic IL-1β level, chronic alcohol abuse might increase the intestinal permeability and gram-negative bacteria in the intestine, which could cause endotoxins to be translocated to the liver through the portal vein combined with receptor complex cluster of differentiation 14 (CD14)-Toll-like receptor 4 (TLR4) in Kupffer’s cells, which then activate the transcription factor, NF-κB, prompting proinflammatory cytokine secretion [58].

In this study, the EL and EC groups had significantly lower plasma ICAM-1 levels compared to the E group (Table 9). The plasma ICAM-1 level is a marker of endothelial activation and vascular inflammation which is correlated with atherosclerosis [59]. Sanadgol et al. reported that palmitic acid could upregulate the expressions of ICAM-1 and VCAM-1 in human bone marrow endothelial cells (HBMECs) pretreated with stimulatory agents such as proinflammatory cytokines or bacterial lipopolysaccharide (LPS), which might play a pivotal role in pathogenesis of cardiovascular events [60]. However, few studies have discussed the effects of SFAs on CVD under a situation of chronic alcohol intake. It remains unclear why plasma ICAM-1 was lower in rats fed lard- and cocoa butt er-containing ethanol diets in this study.

Additionally, the EC and EL groups showed lower hepatic IL-1β, IL-6, IL-10, and TNF-α levels compared to the E group (Table 10), which were comparable to a lower score of inflammatory cell infiltration (Table 3, Figure 1). Multiple regulators are involved in alcohol-induced liver inflammation, including hepatic lipid accumulation, oxidative stress, and endotoxemia [1,2,61,62]. In this study, using lard or cocoa butter as the source of dietary FAs could have decreased hepatic inflammation in the pathohistological examination but caused no changes in hepatic lipid accumulation or oxidative stress (Figure 1, Table 2, Table 4 and Table 5). In an animal study of ethanol-induced liver injury, it was found that using cocoa butter as a dietary fat source increased the protein expression of hepatic argininosuccinate synthase 1 (ASS1) which combines with lipid A (the active position of endotoxin) to inhibit endotoxin activity [63]. Therefore, it was considered that SFAs might be related to inhibition of the endotoxin-liver injury pathway which should be examined in the future.

### 4.6. Correlations between SFAs and Alcoholic Liver Damage

Erythrocytic stearic acid (C18:0) was higher whereas linoleic acid (C18:2) and eicosanoic acid (C20:0) were lower in the E group compared to the C group (Table 11). Based on Table 12, a positive correlation was found between erythrocytic oleic acid (C18:1) and hepatic degeneration and necrosis, but a negative correlation was found for linoleic acid (C18:2). This result is similar to a clinical study which indicated higher oleic acid (C18:1) and lower linoleic acid (C18:2) in serum phospholipids in patients with alcoholic cirrhosis [64]. Erythrocytic linoleic acid (C18:2), arachidonic acid (C20:4), and eicosapentanoic acid (C20:5) were significantly lower in an alcoholic fatty liver and hepatic fibrosis group [65]. Taken together, USFAs, especially oleic acid (C18:1) and linoleic acid (C18:2), appear to be connected to the pathogenesis of alcohol-induced hepatic degeneration and necrosis.

In this study, the EL group had higher palmitic acid (C16:0), palmitoleic acid (C16:1), and steric acid (C18:0), while the EC group had higher steric acid (C18:0) and linoleic acid (C18:2), compared to the E group (Table 11). From this result, it was confirmed that erythrocytic FAs reflect the FA composition in dietary fat (Table 11). The FA composition of red blood cell membranes can represent the whole-body circulation during 2–3 months, while the FA composition in specific cells can reflect the situation of specific tissues [66]. Therefore, it is necessary to analyze the FA composition of hepatocytes in the future.

### 4.7. The Research Limitation

There are some limitations in this study. Estrogen was considered as a pathogenic co-factor of alcoholic liver disease, while the disturbance of the balance of sex hormone in male rats was considered as a pathogenic mechanism for alcoholic liver disease [67]. Further research is needed to consider the effects of different genders on alcoholic liver disease. 

## 5. Conclusions

After 8 weeks of ethanol diet feeding, alcoholic liver injury was successfully established in adult male rats, including higher plasma AST and ALT activities, obvious hepatic damage based on pathohistological examination, higher hepatic TG contents, and higher CYP2E1 protein expression and inflammatory responses (higher plasma ICAM-1 and hepatic IL-1β levels). When cocoa butter or lard was used as the source of dietary fat in an ethanol-containing diet, only cocoa butter inhibited lipid accumulation, however, regardless of whether SFAs were from cocoa butter or lard, hepatic inflammatory cell infiltration, degeneration, and necrosis significantly decreased which might be attributed to lower inflammatory responses, including low plasma ICAM-1 and hepatic IL-1β, IL-6, IL-10, and TNF-α levels. Although dietary lard and cocoa butter did not change the hepatic TG and TC levels or oxidative stress indicators, changes in the plasma lipid profiles should be considered. Moreover, erythrocytic oleic acid (C18:1) was positively and linoleic acid (C18:2) negatively associated with hepatic degeneration and necrosis.

## Figures and Tables

**Figure 1 bioengineering-09-00526-f001:**
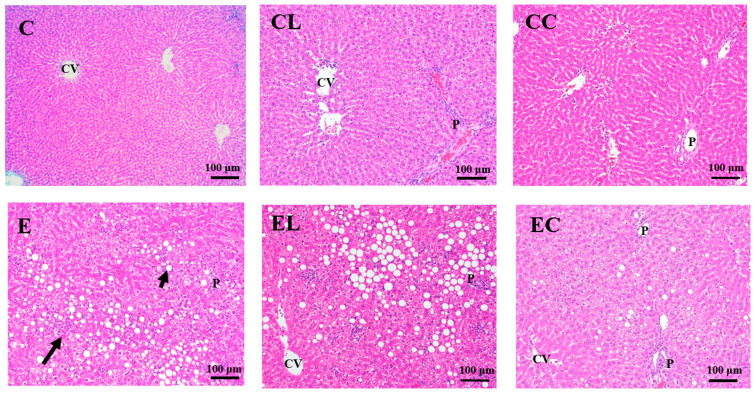
Effects of different dietary fat sources on hematoxylin and eosin (H&E) staining of liver tissue sections in rats. CV, central vein; P, portal area; C, control group; CL, control diet with lard group; CC, control diet with cocoa butter group; E, ethanol group; EL, ethanol diet with lard group; EC, ethanol diet with cocoa butter group. The long arrow shows hepatocyte degeneration and necrosis accompanied by inflammatory cell infiltration. The short arrow indicates lipid droplets.

**Figure 2 bioengineering-09-00526-f002:**
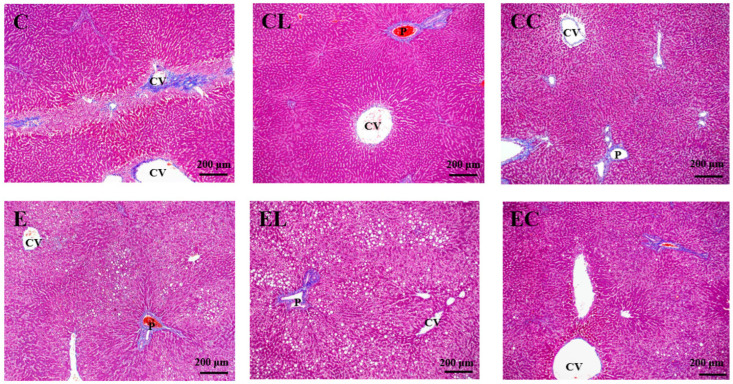
Effects of different dietary fat sources on Masson’s trichrome staining of liver tissue sections in rats. CV, central vein; P, portal area. C, control group; CL, control diet with lard group; CC, control diet with cocoa butter group; E, ethanol group; EL, ethanol diet with lard group; EC, ethanol diet with cocoa butter group.

**Figure 3 bioengineering-09-00526-f003:**
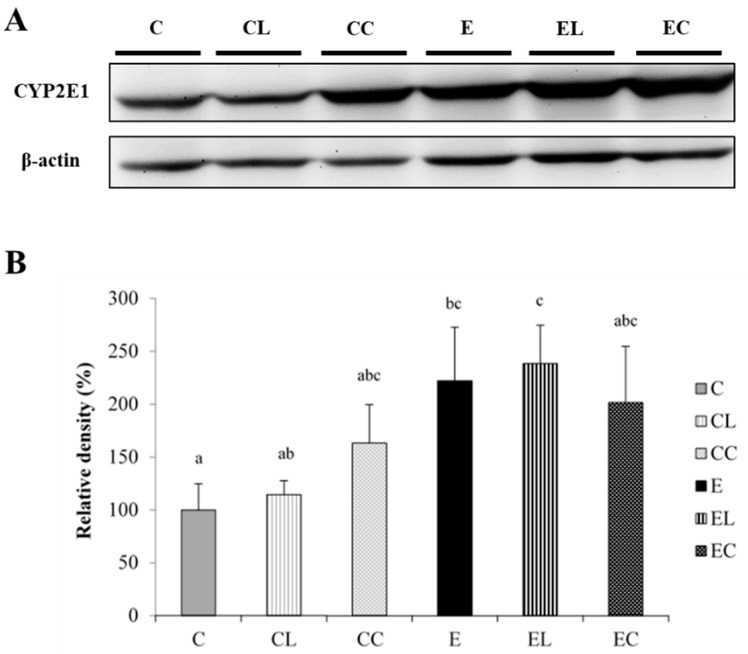
Effects of different dietary fat sources on the protein expression of cytochrome P450 2E1 (CYP2E1) in rats. (**A**) Western blot analysis of CYP2E1 protein expression, with β-actin used as the internal control. (**B**) Quantitative analysis of CYP2E1 levels, and the ratio to β-actin were calculated by setting the value of the C group to 1. Values are expressed as the mean ± SEM (*n* = 6). Bars with different letters significantly differ at the *p* < 0.05 level according to Fisher’s post-hoc test. C, control group; CL, control diet with lard group; CC, control diet with cocoa butter group; E, ethanol group; EL, ethanol diet with lard group; EC, ethanol diet with cocoa butter group.

**Table 1 bioengineering-09-00526-t001:** Composition of the experimental liquid diets.

Component	C	CL	CC	E	EL	EC
	g/L (1000 kcal)					
Casein ^1^	41.4	41.4	41.4	41.4	41.4	41.4
L-Cysteine ^2^	0.5	0.5	0.5	0.5	0.5	0.5
DL-Methionine ^3^	0.3	0.3	0.3	0.3	0.3	0.3
Corn oil ^4^	8.5	0	0	8.5	0	0
Olive oil ^5^	28.4	0	0	28.4	0	0
Safflower oil ^6^	2.7	0	0	2.7	0	0
Lard ^7^	0	35.6	0	0	35.6	0
Cocoa butter ^8^	0	0	35.6	0	0	35.6
Soybean oil ^6^	0	4	4	0	4	4
Choline bitartrate ^9^	0.53	0.53	0.53	0.53	0.53	0.53
Fiber ^10^	10	10	10	10	10	10
Xanthan gum ^11^	4	4	4	4	4	4
ICN: AIN-76 vitamins ^12^	2.5	2.5	2.5	2.5	2.5	2.5
ICN: AIN-76 minerals ^13^	2.6	2.6	2.6	2.6	2.6	2.6
Maltodextrin ^14^	115.2	115.2	115.2	25.6	25.6	25.6
Ethanol ^15^	0	0	0	50	50	50

C, control group; CL, control diet with lard group; CC, control diet with cocoa butter group; E, ethanol group; EL, ethanol diet with lard group; EC, ethanol diet with cocoa butter group. Liquid diet adapted from Lieber and DeCarli. [17]. ^1^ Casein: ICN Biochemicals, 901293; ^2^ L-Cysteine: MP Biomedicals, 101454; ^3^ DL-Methionine: MP Biomedicals, 190955; ^4^ Corn oil: God bene enterprises; ^5^ Olive oil: Standard Foods; ^6^ Safflower oil and soybean oil: Taiwan Sugar; ^7^ Lard: MP Biomedicals, 902140; ^8^ Cocoa butter: MP Biomedicals, 905417; ^9^ Choline bitartrate: MP Biomedicals, 101384; ^10^ Fiber: ICN Biochemicals, 900453; ^11^ Xanthan gum: Sigma-Aldrich, G1253-500G; ^12^ ICN: AIN-76 vitamins: MP Biomedicals, 905454; ^13^ ICN: AIN-76 minerals: MP Biomedicals, 905455; ^14^ Maltodextrin: MP Biomedicals, 960048; ^15^ Ethanol: Sigma-Aldrich, 32205.

**Table 2 bioengineering-09-00526-t002:** Fatty acid composition of the experimental liquid diets ^1^.

Fatty Acids ^2^ (%)	C, E	CL, EL	CC, EC
C14:0	0.5	1.4	0.1
C16:0	11.2	20.3	25.4
C16:1	2.3	5.8	–
C18:0	2.1	10.4	26.7
C18:1 (n-9)	54.5	36.7	32.2
C18:2 (LA, n-6)	27.5	16.6	10.5
C18:3 (ALA, n-3)	0.2	–	–
C20:0	0.6	3.5	2.9
C20:2 (n-6)	0.7	1.4	1.1
C20:3	0.1	–	–
C20:4 (AA, n-6)	–	1.7	0.7
C20:5 (EPA, n-3)	0.3	0.4	0.1
C22:4	0.1	1.2	0.2
C22:5 (DPA, n-3)	–	0.3	0.1
C22:6 (DHA, n-3)	–	0.3	–
SFAs	14.4	35.5	55.1
MUFAs	56.7	42.6	32.2
PUFAs	28.8	21.9	12.7
Total n-3	0.5	1.0	0.2
Total n-6	28.1	19.7	12.3
Total n-9	54.5	36.7	32.2

^1^ C, control group; CL, control diet with lard group; CC, control diet with cocoa butter group; E, ethanol group; EL, ethanol diet with lard group; EC, ethanol diet with cocoa butter group. ^2^ LA, linoleic acid; ALA, alpha-linolenic acid; AA, arachidonic acid; EPA, eicosapentaenoic acid; DPA, docosapentaenoic acid; DHA, docosahexaenoic acid; SFAs, saturated fatty acids; MUFAs, monounsaturated fatty acids; PUFAs, polyunsaturated fatty acids.

**Table 3 bioengineering-09-00526-t003:** Effects of different dietary fat sources on food intake, alcohol intake, body weight (BW), liver weight, and relative liver weight in rats ^1^.

Group ^2^	Food Intake	Alcohol Intake	BW	Relative Liver Weight ^3^
	(g/kg BW/day)	(g/kg BW/day)	(g)	(%)
C	198.5 ± 1.9	-	440.0 ± 5.1^d^	2.3 ± 0.02^a^
CL	197.5 ± 2.4	-	439.4 ± 5.8^d^	2.3 ± 0.04^a^
CC	216.3 ± 3.9	-	384.9 ± 12.5^bc^	2.6 ± 0.10^b^
E	207.4 ± 2.1	10.4 ± 0.1	404.3 ± 11.1^c^	2.9 ± 0.11^c^
EL	206.0 ± 3.9	10.3 ± 0.2	346.3 ± 9.0^a^	3.0 ± 0.08^c^
EC	203.6 ± 2.1	10.2 ± 0.1	363.0 ± 5.1^ab^	2.9 ± 0.05^c^

^1^ Values are expressed as the mean ± SEM. Means in the same column with different superscript letters significantly differ (*p* < 0.05). ^2^ C, control group; CL, control diet with lard group; CC, control diet with cocoa butter group; E, ethanol group; EL, ethanol diet with lard group; EC, ethanol diet with cocoa butter group. ^3^ Relative liver weight: (liver weight/BW) × 100%.

**Table 4 bioengineering-09-00526-t004:** Effects of different dietary fat sources on plasma aspartate transaminase (AST) and alanine transaminase (ALT) activities in rats ^1^.

Group ^2^	AST (U/L)	ALT (U/L)
C	74.5 ± 4.0^a^	36.0 ± 8.7^a^
CL	73.8 ± 6.8^a^	36.0 ± 7.6^a^
CC	870 ± 11.6^a^	51.6 ± 11.4^ab^
E	125.8 ± 23.7^b^	74.6 ± 23.4^c^
EL	176.8 ± 37.1^b^	75.5 ± 46.2^c^
EC	168.5 ± 54.9^b^	70.5 ± 15.7^bc^

^1^ Values are expressed as the mean ± SEM. Means in the same column with different superscript letters significantly differ (*p* < 0.05). ^2^ C, control group; CL, control diet with lard group; CC, control diet with cocoa butter group; E, ethanol group; EL, ethanol diet with lard group; EC, ethanol diet with cocoa butter group.

**Table 5 bioengineering-09-00526-t005:** Effects of different dietary fat sources on hepatic histopathological scores in rats ^1^.

Group ^2^	Fatty Change	Inflammatory Cell Infiltration	Degeneration and Necrosis	Bile Duct Hyperplasia	Fibrosis
C	1.2 ± 0.4^ab^	1.8 ± 0.2^a^	0.2 ± 0.2^a^	0.2 ± 0.2	0.2 ± 0.2
CL	1.6 ± 0.2^abc^	1.8 ± 0.2^a^	0.2 ± 0.2^a^	0.4 ± 0.2	0.0 ± 0.0
CC	0.6 ± 0.4^a^	1.8 ± 0.2^a^	0.2 ± 0.2^a^	0.2 ± 0.2	0.0 ± 0.0
E	2.4 ± 0.2^c^	2.6 ± 0.2^b^	2.8 ± 0.2^c^	0.0 ± 0.0	0.0 ± 0.0
EL	2.4 ± 0.4^c^	1.8 ± 0.2^a^	1.4 ± 0.2^b^	0.0 ± 0.0	0.0 ± 0.0
EC	1.8 ± 0.4^b^	1.8 ± 0.4^a^	1.2 ± 0.2^b^	0.4 ± 0.2	0.4 ± 0.4

^1^ Values are expressed as the mean ± SEM (*n* = 5). Means in the same column with different superscript letters significantly differ (*p* < 0.05). ^2^ C, control group; CL, control diet with lard group; CC, control diet with cocoa butter group; E, ethanol group; EL, ethanol diet with lard group; EC, ethanol diet with cocoa butter group.

**Table 6 bioengineering-09-00526-t006:** Effects of different dietary fat sources on blood lipid profiles in rats ^1,2^.

Group ^3^	TGs	TC	HDL-C	LDL-C	TC/HDL-C Ratio
	(mg/dL)	(mg/dL)	(mg/dL)	(mg/dL)	
C	50.5 ± 4.0	52.5 ± 3.9^ab^	12.9 ± 0.9^a^	6.6 ± 0.6^b^	4.1 ± 0.1^bc^
CL	41.4 ± 5.9	51.9 ± 2.9^a^	12.3 ± 0.5^a^	7.1 ± 0.3^b^	4.2 ± 0.1^c^
CC	47.2 ± 4.2	52.5 ± 2.4^ab^	12.5 ± 1.0^a^	7.7 ± 0.7^b^	4.4 ± 0.1^c^
E	37.2 ± 3.9	63.3 ± 4.2^b^	14.6 ± 0.9^a^	4.2 ± 0.4^a^	4.3 ± 0.1^c^
EL	60.4 ± 21.5	77.7 ± 4.4^c^	20.9 ± 0.7^b^	7.1 ± 1.1^b^	3.7 ± 0.2^ab^
EC	52.4 ± 5.6	76.2 ± 5.6^c^	20.3 ± 0.9^b^	7.6 ± 1.3^b^	3.7 ± 0.1^ab^

^1^ Values are expressed as the mean ± SEM. Means in the same column with different superscript letters significantly differ (*p* < 0.05). ^2^ TGs, triglycerides; TC, total cholesterol; HDL-C, high-density lipoprotein cholesterol; LDL-C, low-density lipoprotein cholesterol. ^3^ C, control group; CL, control diet with lard group; CC, control diet with cocoa butter group; E, ethanol group; EL, ethanol diet with lard group; EC, ethanol diet with cocoa butter group.

**Table 7 bioengineering-09-00526-t007:** Effects of different dietary fat sources on hepatic triglycerides (TGs) and total cholesterol (TC) concentrations in rats ^1^.

Group ^2^	TGs	TC
	(mg/g Liver)	(mg/g Liver)
C	14.1 ± 1.6^ab^	21.2 ± 0.6
CL	15.5 ± 1.2^bc^	26.3 ± 5.6
CC	13.2 ± 1.2^b^	20.9 ± 0.9
E	18.9 ± 1.5^c^	21.9 ± 1.4
EL	17.0 ± 1.1^bc^	23.3 ± 1.1
EC	18.0 ± 2.2^bc^	19.8 ± 0.9

^1^ Values are expressed as the mean ± SEM. Means in the same column with different superscript letters significantly differ (*p* < 0.05). ^2^ C, control group; CL, control diet with lard group; CC, control diet with cocoa butter group; E, ethanol group; EL, ethanol diet with lard group; EC, ethanol diet with cocoa butter group. As shown in Table 2.

**Table 8 bioengineering-09-00526-t008:** Effects of different dietary fat sources on the antioxidant status in rats ^1,2^.

Group ^3^	Hepatic GSH/GSSG Ratio	Hepatic TBARSs (μM)
C	6.0 ± 0.9^ab^	10.6 ± 0.6
CL	8.9 ± 0.8^bc^	10.9 ± 1.1
CC	7.6 ± 2.6^bc^	12.1 ± 1.3
E	5.1 ± 0.3^a^	13.6 ± 1.3
EL	5.4 ± 0.4^ab^	14.8 ± 2.1
EC	6.5 ± 0.7^ab^	13.5 ± 3.8

^1^ Values are expressed as the mean ± SEM. Means in the same column with different superscript letters significantly differ (*p* < 0.05). ^2^ GSH, reduced glutathione; GSSG, oxidized glutathione; TBARSs, thiobarbituric acid-reactive substances. ^3^ C, control group; CL, control diet with lard group; CC, control diet with cocoa butter group; E, ethanol group; EL, ethanol diet with lard group; EC, ethanol diet with cocoa butter group.

**Table 9 bioengineering-09-00526-t009:** Effects of different dietary fat sources on plasma vascular cell adhesion molecule (VCAM)- 1, intercellular adhesion molecule (ICAM)-1, and E-selectin levels in rats ^1^.

Group ^2^	VCAM-1	ICAM-1	E-Selectin
	(ng/mL)	(ng/mL)	(ng/mL)
C	190.9 ± 20.7^ab^	13.6 ± 0.6^a^	24.7 ± 1.3
CL	168.8 ± 22.0^ab^	12.8 ± 0.7^a^	23.3 ± 1.3
CC	161.4 ± 27.3^a^	11.6 ± 0.4^a^	23.0 ± 1.8
E	286.7 ± 80.6^ab^	17.8 ± 2.6^b^	25.0 ± 2.1
EL	307.5 ± 61.0^b^	13.3 ± 1.1^a^	20.6 ± 1.6
EC	276.9 ± 40.9^ab^	13.1 ± 0.8^a^	21.4 ± 1.4

^1^ Values are expressed as the mean ± SEM. Means in the same column with different superscript letters significantly differ (*p* < 0.05). ^2^ C, control group; CL, control diet with lard group; CC, control diet with cocoa butter group; E, ethanol group; EL, ethanol diet with lard group; EC, ethanol diet with cocoa butter group.

**Table 10 bioengineering-09-00526-t010:** Effects of different dietary fat sources on hepatic interleukin (IL)-1β, IL-6, IL-10, and tumor necrosis factor (TNF)-α levels in rats ^1^.

Group ^2^	TNF-α	IL-1β	IL-6	IL-10
	(pg/mg Protein)	(pg/mg Protein)	(pg/mg Protein)	(pg/mg Protein)
C	0.4 ± 0.03^b^	5.8 ± 0.6^a^	1.7 ± 0.1^bc^	6.2 ± 0.6^bc^
CL	0.4 ± 0.03^b^	5.9 ± 0.7^a^	1.6 ± 0.1^bc^	6.3 ± 0.5^bc^
CC	0.4 ± 0.03^b^	4.8 ± 0.5^a^	1.4 ± 0.1^b^	6.5 ± 0.7^c^
E	0.4 ± 0.02^b^	9.6 ± 2.3^b^	1.9 ± 0.3^c^	4.9 ± 0.3^b^
EL	0.2 ± 0.01^a^	3.8 ± 0.5^a^	0.8 ± 0.1^a^	3.1 ± 0.2^a^
EC	0.2 ± 0.02^a^	3.4 ± 0.5^a^	0.7 ± 0.1^a^	3.3 ± 0.4^a^

^1^ Values are expressed as the mean ± SEM. Means in the same column with different superscript letters significantly differ (*p* < 0.05). ^2^ C, control group; CL, control diet with lard group; CC, control diet with cocoa butter group; E, ethanol group; EL, ethanol diet with lard group; EC, ethanol diet with cocoa butter group.

**Table 11 bioengineering-09-00526-t011:** Effects of different dietary fat sources on erythrocytic fatty acid composition in rats ^1^.

Fatty Acid ^2^ (%)	C	CL	CC	E	EL	EC
C14:0	1.90 ± 0.26^a^	2.10 ± 0.31^a^	2.00 ± 0.33^a^	2.00 ± 0.31^a^	3.10 ± 0.31^b^	2.00 ± 0.25^a^
C16:0	41.1 ± 1.00^bcd^	43.5 ± 0.94^d^	40.1 ± 0.88^abc^	39.1 ± 0.28^ab^	42.4 ± 0.55^cd^	38.0 ± 0.97^a^
C16:1	0.40 ± 0.10^ab^	0.50 ± 0.08^bc^	0.30 ± 0.09^ab^	0.40 ± 0.03^b^	0.70 ± 0.14^c^	0.20 ± 0.07^a^
C18:0	18.6 ± 0.84^a^	20.2 ± 0.82^ab^	24.3 ± 0.52^d^	22.2 ± 0.32^c^	21.8 ± 0.80^bc^	26.0 ± 0.55^d^
C18:1 (n-9)	15.6 ± 1.07^bc^	13.4 ± 0.36^a^	14.4 ± 0.29^ab^	16.8 ± 0.40^c^	16.4 ± 0.83^c^	15.5 ± 0.49^bc^
C18:2 (LA, n-6)	8.30 ± 0.52^cd^	8.80 ± 0.49^d^	6.80 ± 0.31^ab^	6.10 ± 0.17^a^	7.40 ± 0.39^bc^	7.20 ± 0.16^b^
C18:3 (ALA, n-3)	–	0.10 ± 0.15	–	–	–	–
C20:0	1.60 ± 0.31^b^	1.20 ± 0.25^b^	0.50 ± 0.03^a^	0.50 ± 0.04^a^	0.30 ± 0.10^a^	0.30 ± 0.06^a^
C20:2 (n-6)	2.90 ± 0.83^ab^	1.30 ± 0.46^a^	1.80 ± 0.63^ab^	3.20 ± 0.29^b^	1.40 ± 0.59^a^	1.70 ± 0.66^ab^
C20:3	0.70 ± 0.29	0.90 ± 0.54	0.50 ± 0.25	1.20 ± 0.15	0.30 ± 0.26	0.30 ± 0.26
C20:4 (AA, n-6)	6.20 ± 1.51^ab^	5.70 ± 0.88^ab^	6.40 ± 1.00^b^	5.60 ± 0.31^ab^	3.50 ± 0.70^a^	5.70 ± 1.04^ab^
C20:5 (EPA, n-3)	0.60 ± 0.27^b^	0.10 ± 0.04^a^	0.10 ± 0.04^a^	0.20 ± 0.04^ab^	0.30 ± 0.15^ab^	0.10 ± 0.04^a^
C22:0	0.20 ± 0.22	–	–	–	0.10 ± 0.11	0.40 ± 0.29
C22:4	1.30 ± 0.53	1.40 ± 0.34	2.10 ± 0.59	2.00 ± 0.08	1.60 ± 0.54	0.70 ± 0.43
C22:5 (DPA, n-3)	0.40 ± 0.07	0.40 ± 0.09	0.40 ± 0.07	0.20 ± 0.01	0.30 ± 0.04	0.40 ± 0.07
C22:6 (DHA, n-3)	0.30 ± 0.14	0.40 ± 0.16	0.50 ± 0.18	0.30 ± 0.03	0.40 ± 0.15	0.60 ± 0.15
SFAs	63.3 ± 1.53^a^	67.0 ± 0.91^bc^	66.8 ± 1.20^bc^	63.9 ± 0.39^ab^	67.8 ± 1.02^c^	67.6 ± 1.37^c^
MUFAs	15.9 ± 1.05^bc^	13.9 ± 0.42^a^	14.7 ± 0.37^ab^	17.2 ± 0.41^c^	17.1 ± 0.87^c^	15.7 ± 0.53^abc^
PUFAs	20.8 ± 1.66^c^	19.2 ± 0.79^bc^	18.5 ± 1.23^bc^	18.9 ± 0.19^bc^	15.1 ± 0.59^a^	16.7 ± 1.79^ab^
Total n-3	1.30 ± 0.33	1.00 ± 0.18	0.90 ± 0.27	0.80 ± 0.05	0.90 ± 0.19	1.10 ± 0.21
Total n-6	17.4 ± 1.24^b^	15.9 ± 0.66^b^	15.0 ± 0.96^ab^	14.9 ± 0.30^ab^	12.2 ± 0.43^a^	14.6 ± 1.60^ab^
Total n-9	15.6 ± 1.07^bc^	13.4 ± 0.36^a^	14.4 ± 0.29^ab^	16.8 ± 0.40^c^	16.4 ± 0.83^c^	15.5 ± 0.49^bc^
SFAs/MUFAs	4.10 ± 0.30^ab^	4.80 ± 0.19^c^	4.60 ± 0.15^bc^	3.70 ± 0.11^a^	4.00 ± 0.26^ab^	4.30 ± 0.10^bc^
SFAs/PUFAs	3.10 ± 0.30^a^	3.50 ± 0.19^ab^	3.70 ± 0.28^abc^	3.40 ± 0.04^a^	4.50 ± 0.22^c^	4.30 ± 0.52^bc^
MUFAs/PUFAs	0.80 ± 0.11^ab^	0.70 ± 0.04^a^	0.80 ± 0.06^ab^	0.90 ± 0.03^abc^	1.10 ± 0.08^c^	1.00 ± 0.13^bc^
SFAs/USFAs	1.70 ± 0.11^a^	2.00 ± 0.09^bc^	2.00 ± 0.10^abc^	1.80 ± 0.03^ab^	2.10 ± 0.10^c^	2.10 ± 0.13^c^
n-6/n-3	19.5 ± 6.94	17.5 ± 2.31	31.6 ± 14.0	18.8 ± 1.39	18.3 ± 7.03	15.7 ± 3.21
n-9/n-3	20.4 ± 10.3	14.6 ± 1.64	32.5 ± 15.1	21.1 ± 1.55	23.6 ± 8.15	19.6 ± 7.27
n-9/n-6	0.90 ± 0.11^a^	0.90 ± 0.04^a^	1.00 ± 0.06^a^	1.10 ± 0.04^ab^	1.30 ± 0.09^b^	1.10 ± 0.17^ab^

^1^ Values are expressed as the mean ± SEM (*n* = 5). Means in the same column with different superscript letters significantly differ (*p* < 0.05). C, control group; CL, control diet with lard group; CC, control diet with cocoa butter group; E, ethanol group; EL, ethanol diet with lard group; EC, ethanol diet with cocoa butter group. ^2^ LA, linoleic acid; ALA, alpha-linolenic acid; AA, arachidonic acid; EPA, eicosapentaenoic acid; DPA, docosapentaenoic acid; DHA, docosahexaenoic acid; SFAs, saturated fatty acids; MUFAs, monounsaturated fatty acids; PUFAs, polyunsaturated fatty acids; USFAs, unsaturated fatty acids.

**Table 12 bioengineering-09-00526-t012:** The correlation of erythrocytic fatty acid composition and hepatic histopathological analysis scores in rats.

Fatty Acids ^1^	Inflammatory Cell Infiltration		Degeneration and Necrosis		Fatty Change	
	*r*	*p* Value	*r*	*p* Value	*r*	*p* Value
C14:0	0.149	0.544	0.151	0.538	0.556	0.513
C16:0	0.020	0.937	−0.272	0.260	−0.159	0.516
C16:1	0.200	0.411	0.054	0.827	0.250	0.303
C18:0	−0.105	0.669	0.096	0.695	0.232	0.340
C18:1 (n-9)	0.359	0.131	0.667	0.002 *	0.218	0.370
C18:2 (LA, n-6)	−0.167	0.493	−0.565	0.012 *	−0.287	0.233
C20:0	−0.117	0.633	−0.439	0.060	-0.384	0.105
C20:2 (n-6)	−0.147	0.547	−0.252	0.298	−0.118	0.631
C20:3	0.353	0.138	0.121	0.622	0.044	0.860
C20:4 (AA, n-6)	−0.108	0.661	0	0.987	−0.088	0.719
C20:5 (EPA, n-3)	0.030	0.906	−0.201	0.409	−0.224	0.356
C22:0	−0.330	0.167	0.042	0.864	-0.171	0.484
C22:4	0.129	0.599	0.345	0.148	0.050	0.838
C22:5 (DPA, n-3)	−0.202	0.408	−0.377	0.111	0.017	0.943
C22:6 (DHA, n-3)	−0.179	0.465	−0.091	0.710	−0.085	0.729
SFAs	−0.127	0.604	−0.190	0.436	0.081	0.742
MUFAs	0.385	0.104	0.681	0.001 *	0.240	0.323
PUFAs	−0.096	0.697	−0.207	0.395	−0.245	0.313
Total n-3	−0.144	0.556	−0.307	0.201	−0.219	0.368
Total n-6	−0.251	0.299	−0.387	0.102	−0.276	0.253
Total n-9	0.359	0.131	0.667	0.002 *	0.218	0.370
SFAs/MUFAs	−0.307	0.201	−0.587	0.008 *	−0.178	0.465
SFAs/PUFAs	−0.028	0.907	0.090	0.715	0.247	0.307
MUFAs/PUFAs	0.210	0.389	0.465	0.045 *	0.353	0.138
SFAs/USFAs	−0.155	0.527	−0.221	0.364	0.119	0.628
n-6/n-3	−0.281	0.245	0.041	0.865	−0.162	0.507
n-9/n-3	−0.199	0.414	0.080	0.746	−0.071	0.771
n-9/n-6	0.353	0.139	0.687	0.001 *	0.271	0.261

* Correlation is significant (*p* < 0.05). ^1^ LA, linoleic acid; ALA, alpha-linolenic acid; AA, arachidonic acid; EPA, eicosapentaenoic acid; DPA, docosapentaenoic acid; DHA, docosahexaenoic acid; SFAs, saturated fatty acids; MUFAs, monounsaturated fatty acids; PUFAs, polyunsaturated fatty acids; USFAs, unsaturated fatty acids.

## Data Availability

The data presented in this study are available on request from the corresponding author.

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
