# Peer review of "The Preliminary Results for Evaluating Cocoa Butter’s Hepatoprotective Effects against Lipid Accumulation and Inflammation in Adult Male Rats Chronically Fed Ethanol"

_bioengineering, 2022, doi:10.3390/bioengineering9100526_

Round 1

Reviewer 1 Report

The manuscript is well written in a good English. The technical aspects of the presented data seem to be good, and the experimental work is properly and accurately performed. In the discussion section the authors provide a good presentation of their results. In my opinion the manuscript is suitable for the publication on Bioengineering in its present form.

Author Response

Your encouragement is greatly appreciated. In the future, we will continue to conduct excellent research.

Reviewer 2 Report

Review of the paper bioengineering-1939225 submitted for publication in Bioengineering. The paper titled “Cocoa butter protects the liver against lipid accumulation and inflammation in rats chronically fed ethanol” aimed to clarify the role of saturated fats from cocoa butter (plant source) compared with lard (animal source) on alcoholic liver damage in rats.

The study used several basic biochemical parameters to investigate the effects under 6 conditions these being C, control group; CL, control diet with lard group; CC, control diet with cocoa butter group; E, ethanol group; EL, ethanol diet with lard group; EC, ethanol diet with cocoa butter group.

Unfortunately the authors did not consider gender effect in this study, which can be the main drawback. The conclusions should be then focused on males only. Also, the authors used only aged 8-week-old male Wistar rats, thus the age was not considered. This is another weakness of the study to drive robust conclusions.

Considering the above the title should be changed by adding two things “specify the gender” and mention “preliminary results”.

There are several tables in this papers and they are not all needed. Please revise and move few of them to the supplementary and keep only what is important and hugely discussed in the manuscript.

The figure 4 is confusing and not adding anything to the paper. It should be removed.

The discussion is long, not focused. It should be improved. The authors need to focus on the factors of the study with more discussion about males. Avoid using other species as the mechanisms can be very different.

Author Response

  1. Unfortunately the authors did not consider gender effect in this study, which can be the main drawback. The conclusions should be then focused on males only. Also, the authors used only aged 8-week-old male Wistar rats, thus the age was not considered. This is another weakness of the study to drive robust conclusions. Considering the above the title should be changed by adding two things “specify the gender” and mention “preliminary results”.

Response: Thank you for the great suggestion. As the reviewer said, gender is always the problem when a researcher tries to give the conclusion for the animal study. The prejudice against using female animals may be partly due to concerns that they are intrinsically more variable than males because of cyclical reproductive hormones, making them unsuitable for use as baseline models (Wang, 1923). However, Zucker and Beery indicated that Journal editors and reviewers should require authors of research studies that use only male or only female animals to state this in the title of their papers (Zucker and Beery, 2010). Therefore, we have revised the title as the reviewer’s consideration. The revised title is “The preliminary results for evaluating cocoa butter’s hepatoprotective effects against lipid accumulation and inflammation in adult male rats chronically fed ethanol”. Furthermore, we also added some words to indicate the results were obtained from adult male rats in the conclusion (Line 536).

Wang, G. H. Relation between “spontaneous” activity and oestrous cycle in the white rat. Comp. Psychol. Monog. 6, 1-27 (1923).

Zucker I, Beery AK. Males still dominate animal studies. Nature. 465, 690 (2010).

  1. There are several tables in this paper, and they are not all needed. Please revise and move few of them to the supplementary and keep only what is important and hugely discussed in the manuscript. The figure 4 is confusing and not adding anything to the paper. It should be removed.

Response: Thank you for the comment. We have deleted figure 4. As part of the writing process, we usually use diagrams to illustrate the conclusions. However, the figure can be lifted as the reviewer’s consideration.

  1. The discussion is long, not focused. It should be improved. The authors need to focus on the factors of the study with more discussion about males. Avoid using other species as the mechanisms can be very different.

Response: Thank you for the comment. We discussed whether alcoholic liver damage was successfully established or not such as 3.1 and 3.2. Moreover, we discussed the pathological mechanisms, including lipid metabolism (3.3), oxidative stress (3.4) and inflammation (3.5). Then, we discussed the correlation between fatty acid composition and alcoholic liver damage (3.6) which is the most important part to explain the relationship between dietary fat source and alcohol intake. As last, we added the research limitation (Line 416-422) to describe the sex bias as the reviewer’s comments. Hopefully, the revised version will meet the reviewer's requirements.
